🔓 | **Open Peer Review** | Human Microbiome | Research Article

# Role of age in mediating the association between the vaginal microbiota and preterm birth

Yun Xie,[1] Qi Wang,[2] Dan Li,[1] Fengan Jia,[3] Fan Chang,[3] Zhen Zhang,[4] Yanmei Sun,[4] Shiwei Wang[4]

**ABSTRACT**   The vaginal microbiota plays a crucial role in vaginal health and pregnancy outcomes. However, the influence of maternal characteristics, especially age, on the relationship between the vaginal microbiota and preterm birth is poorly understood. This study quantifies the effects of race, maternal age, and gestational age on vaginal microbiota composition using publicly available 16S rRNA sequencing data. Our results demonstrate stronger associations between preterm birth and a less optimal vaginal microbiota in White and Asian women compared to Black or African American women. The microbiota difference between preterm and term births was more pronounced in older reproductive-age women, with *Lactobacillus* species increasing with age only in term births. Additionally, lower alpha diversity was observed in term pregnancies compared to preterm during both early and late pregnancy, particularly in women aged above 25 years. These findings highlight the mediating role of age in the relationship between the vaginal microbiota and preterm birth.

**IMPORTANCE** The findings from this study have potential clinical implications for managing preterm birth (PTB) risks. By identifying maternal age as a key factor affecting the vaginal microbiota, healthcare providers can develop age-specific preventive strategies. Interventions aimed at optimizing the vaginal microbiota in older reproductive-age women may be more effective than in younger women for reducing PTB risk. These insights could inform personalized clinical approaches, ultimately improving pregnancy outcomes.

**KEYWORDS**   vaginal microbiota, vaginal microbiome, preterm, age, race, gestational age

The vaginal microbiota (VMB) is a critical component of the female reproductive system, playing an essential role in maintaining vaginal health and influencing pregnancy outcomes (1). The dominance of *Lactobacillus* species in the VMB is a key indicator of overall vaginal health. Protective *Lactobacillus* contributes to microbial balance by producing lactic acid, which helps maintain a low vaginal pH, suppresses the growth of pathogens, and supports immune modulation. In contrast, many non-*Lactobacillus* species, such as *Gardnerella vaginalis* and *Sneathia amnii*, produce virulence factors and are linked to various diseases in the female lower reproductive tract.

Preterm birth (PTB), defined as delivery before 37 weeks of gestation, is a major global health concern, accounting for approximately 10% of all live births and contributing to high rates of neonatal morbidity and mortality (2). The causes of PTB are multifactorial, involving genetic, environmental, and microbial factors (3). Among these, the role of the VMB has received significant attention. Studies have shown that certain bacterial taxa are associated with an increased risk of PTB, including *G. vaginalis*, *S. amnii*, and other dysbiosis-related species (1). Conversely, a VMB dominated by health-associated *Lactobacillus* species, e.g., *L. crispatus*, *L. jensenii*, and *L. gasseri*, is generally linked to

Address correspondence to Shiwei Wang, wangsw@nwu.edu.cn.

The authors declare no conflict of interest.

See the funding table on p. 14.

better pregnancy outcomes. *L. iners* is the only *Lactobacillus* species whose role in overall vaginal health remains unclear (4). Recent studies suggest that a vaginal microbiome dominated by *L. iners* may represent an intermediate or transitional state of microbial composition (5).

The composition of the VMB is influenced by several factors, including race and gestational stage (1). Studies have consistently reported that Black/African American (B/AA) women are more likely to have VMBs characterized by higher microbial diversity and increased prevalence of dysbiosis-associated taxa compared to White women (6). Gestational age also plays a critical role, as during pregnancy, the VMB undergoes substantial compositional and functional changes, marked by an increased dominance of *Lactobacillus* species and a reduction in taxa associated with dysbiosis (7). This change is thought to represent a protective mechanism that supports healthier delivery outcomes.

Despite advances in VMB research, the interaction between the VMB and maternal characteristics in pregnancy outcomes remains unclear. Most studies on VMB and PTB are cross-sectional and often neglect confounding factors such as race and gestational age (8). Additionally, studies on the association between maternal age and the VMB primarily compare the VMB in postmenopausal and reproductive-age women (9), leaving the role of maternal age in the association between the VMB of reproductive-age women and PTB poorly understood.

In this study, we sought to address these gaps by analyzing a large, high-quality data set of VMB samples collected from pregnant women across diverse racial groups, ages, and gestational ages. Our data elucidate the influence of race, age, and gestational age on the association between the VMB and PTB by integrating data from multiple cohorts and employing robust statistical approaches. Our findings underscore more pronounced differences in the VMB between PTB and term births in older women of reproductive age. The data also highlight that, in term births, the relative abundances of *L. crispatus* and *L. jensenii* increase with age, while the abundance of dysbiosis-associated taxa decreases, but similar trends are not seen in PTBs.

## MATERIALS AND METHODS

### Inclusion and exclusion criteria

Publications were searched in the PubMed database using a strategy that incorporated three key terms: ("vaginal microbiota" OR "vaginal microbiome") AND (16S rRNA) AND (preterm OR premature). The inclusion criteria were as follows: (i) raw 16S rRNA sequencing data were publicly available for download from the Sequence Read Archive (SRA; https://www.ncbi.nlm.nih.gov/sra); (ii) the metadata of participants, i.e., race, age, gestational age, and pregnancy outcome, were clearly specified; and (iii) the study was an original research article. The following exclusion criteria were included: (i) studies not related to the VMB, (ii) studies with fewer than 30 samples, and (iii) participants under the age of 16 years were excluded.

### Data preprocessing

Following quality control, paired sequence reads were trimmed and merged, and human-origin reads were removed as outlined previously (10). The resulting high-quality 16S rRNA amplicon sequences were then aligned to a refined 16S rRNA database described in the earlier research (10). Taxa were included in the feature table only if their relative abundance reached at least 0.1% (or 0.01%) in 5% (or 15%) of the samples, as previously reported (10). Samples with fewer than 5,000 total reads were excluded from further analysis.

## Removal of batch effect

Batch effects across cohorts were corrected using the sva package in R. Specifically, the feature table of 16S rRNA profiles was normalized using centered log-ratio (CLR) normalization, and the ComBat function was applied to address batch effects.

## Distance-based redundancy analysis

The influence of multiple variables, i.e., race, age, gestational age, and pregnancy outcome, on the composition of the VMB was assessed using the distance-based redundancy analysis (dbRDA), implemented through the capscale function in R with the parameter distance set to "bray" (11). Statistical significance was evaluated using the marginal Adonis test, with the parameter "by" set to "margin." To ensure comparable degrees of freedom among these variables, age was categorized into three levels: "low," "medium," and "high," with cases evenly distributed across these groups. Similarly, gestational age was recoded into three categories.

## Alpha and beta diversity

The feature table derived from 16S rRNA profiles was normalized through rarefaction, matching the sequencing depth of the sample with the fewest reads (>5,000). To quantify alpha diversity, the Shannon index was computed using the R package vegan (11), and group differences were analyzed via the two-sided Wilcoxon test. Alternatively, in case small sample sizes in certain subgroup analyses increase the risk of spurious associations, the Shannon index was adjusted by the inverse probability weighting procedure. More specifically, propensity scores were estimated using a logistic regression model with the glm function in R, where the probability of the outcome (PTB vs term birth) was predicted based on potential confounders, i.e., gestational age at collection and maternal age. Next, inverse probability weights were computed using these scores to balance the data. The weights were applied in the weighted regression using the svyglm function of the survey package in R. This approach ensured that the association between the Shannon index and pregnancy outcome was not biased by confounding factors. Beta diversity was visualized using the prcomp function in R to calculate the principal coordinate analysis (PCoA) values. Group-level beta diversity differences were examined througha permutational multivariate analysis of variance (PERMANOVA) analysis using the "adonis2" function provided in the vegan package.

## Differential abundance analysis

Differential abundance analysis was performed using the ALDEx2 package in R (12). The adjusted *P* values for differences in relative abundance were computed with the aldex.ttest function, employing the Mann-Whitney U test and subsequent Benjamini-Hochberg correction. Changes in relative abundance were quantified using the aldex.effect function, which calculated the median difference per feature between two conditions.

## Interaction between age and preterm birth in relation to bacterial taxa

A mixed-effects model was implemented using the lm function from the lme4 package, with the formula specified as follows: bacterium ~ age × pregnancy outcome. The bacterial abundances were the CLR normalized values of bacterial reads. Two sets of *P* values were generated: one for the correlation between age and bacterial abundance, and another for the interaction effect between age and pregnant outcomes on bacterial abundance. Both sets of *P* values were adjusted using the Benjamini-Hochberg correction to control for multiple comparisons.

## RESULTS

### Overall profiles of the vaginal microbiota

This study analyzed 2,408 samples (Fig. S1) from four BioProjects (Fig. S2a) (13–16), selected based on criteria described in Materials and Methods. Alpha rarefaction analysis determined an appropriate threshold for total reads, showing no significant difference in observed taxa between rarefaction depths of 1,600 and 1,800 reads (Fig. S1). To ensure high-quality data, samples with fewer than 5,000 reads were excluded as previously described (10), leaving 2,266 samples for analysis. Low-abundance and rarely observed taxa were also removed, resulting in 143 species-level taxa for further study.

The final data set included samples mostly from the United States (Fig. S2b). These included 516 PTB and 1,750 term-birth participants (Fig. S2c). The majority of participants were B/AA, followed by White and smaller proportions of Asian and other racial groups (Fig. S2d). The average gestational age was $160.2 \pm 60.8$ days, and the mean participant age was $28.2 \pm 5.3$ years (Fig. S2e and f). Batch effect correction was applied to ensure data comparability across cohorts (Fig. S3).

Using all samples without case matching, alpha diversity (Shannon index) was significantly lower in term-birth VMBs (Fig. S4a). Beta diversity, visualized with PCoA and quantified using the Adonis test, indicated a significant association between VMB composition and PTB (Fig. S4b). Differential abundance analysis identified five *Lactobacillus* species, e.g., *L. crispatus* and *L. jensenii*, as more abundant in term-birth VMBs (Fig. S4c). Dysbiosis-associated taxa, e.g., *S. amnii*, *Lachnospiraceae* BVAB1, and *G. vaginalis*, were more abundant in preterm VMBs. These findings align with prior studies (1).

### dbRDA on factors associated with preterm

The associations between the VMB and factors, i.e., race, PTB, and gestational age, have been explored (1), but their relative impacts on VMB composition have been quantified by limited studies (17). Additionally, the relationship between age and the VMB in reproductive-aged individuals has not been well defined. To address these gaps, distance-based redundancy analysis was applied to visualize the effects of these factors on VMB composition, and an Adonis test with a marginal model was used to quantify their impact. For comparability, age (Fig. 1a) and gestational age (Fig. 1b) were divided into three categories with similar case numbers to achieve a comparable degree of freedom to race and pregnancy outcome in the dbRDA. This approach also helps to prevent bias in group size distribution.

Results of the Adonis test with a marginal model showed that race, age, gestational age, and birth outcome all significantly influenced VMB composition (Fig. 1c). Race accounted for the most variance, followed by age, gestational age, and birth outcome. VMBs from women aged 25.1–32 years, White women, or those with term births were enriched in *L. crispatus* and *L. gasseri* (Fig. 1d; Data set S1). Conversely, VMBs from women aged 17–25 years or of B/AA race were enriched in *L. iners* and *Lachnospiraceae* BVAB1. Samples collected earlier in pregnancy (arrow labeled "Gestational_age: Low" in Fig. 1d) showed higher abundances of *Atopobium vaginae* and *G. vaginalis*. These findings align with previous studies showing that White women have VMBs enriched with health-associated *Lactobacillus* species compared to B/AA women (6, 15) and that the VMB tends to enrich in *Lactobacillus* species and reduce dysbiosis-associated taxa during pregnancy (1, 7). Age was identified as the second most influential factor, with women aged 25.1–32 years having more optimal VMB profiles.

### Association between the vaginal microbiota and preterm birth in different races

Some previous studies on the association between the VMB and PTB have been cross-sectional with limited control for confounding factors (8). Although some studies matched samples based on race or ethnicity, age, and household income (18), they

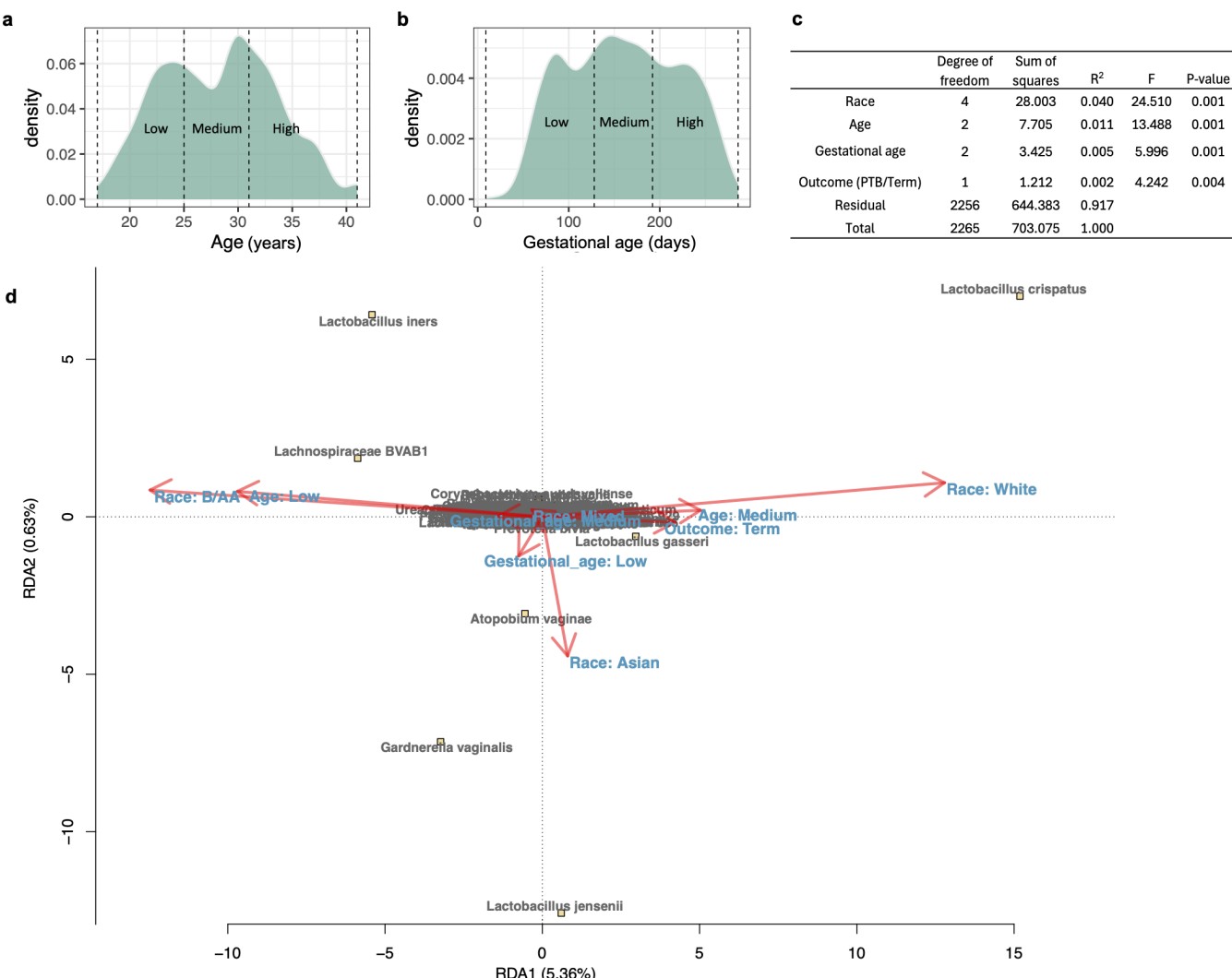

**FIG 1** The composition of the VMB influenced by maternal factors. Age (a) and gestational age (b) divided into three categories, respectively, based on evenly distributed case numbers are shown. (c) Results of the Adonis test using a marginal model to quantify the effects of age, gestational age, and pregnancy outcome on VMB composition are illustrated. PTB represents preterm birth. (d) dbRDA evaluating the influence of maternal factors on the composition of the VMB is shown. Detailed coordinates of bacterial taxa on the dbRDA plot are listed in Data set S1.

lacked in-depth analyses of specific subgroups classified by race, maternal age, and gestational age. Here, VMBs from PTB and term-birth cases were matched based on race, BioProject, and similar age and gestational age (Fig. S4d). The results showed that both alpha (Fig. S4e) and beta (Fig. S4f) diversities of the VMB remained significantly associated with PTB, but the significance level was obviously lower than in the unmatched analysis (Fig. S4a and b). Additionally, no taxa showed significant differences in relative abundance between PTB and term-birth cases. These findings, together with the dbRDA, suggest that race, age, and gestational age are confounding factors in studying the association between the VMB and PTB.

To exclude the impact of race, analyses were performed within specific racial groups. For each comparison, samples were matched by BioProject, age, and gestational age (Fig. S5a through c). The results showed that VMBs from White and Asian women, but not B/AA women, had significantly lower alpha diversity in term births compared to PTB (Fig. 2a). Consistent with previous research (17), the beta diversity association with PTB was also stronger in White and Asian women than in B/AA women (Fig. 2b). In B/AA women, preterm birth was associated with higher levels of *Prevotella buccalis*, while term

births in White women were enriched in *L. crispatus*, *L. jensenii*, and *L. kitasatonis* (Fig. 2c). In Asian women, term births showed higher levels of *L. crispatus*, *L. jensenii*, and *L. iners*, while preterm births had increased dysbiosis-related taxa, such as *G. vaginalis* and *Streptococcus anginosus*. However, most Asian participants were aged 32.1–42, which may have influenced these results, as discussed further below.

Further analysis of B/AA women using case-matched VMBs (Fig. S5d and e) from two BioProjects, PRJNA393472 and PRJNA725416, showed lower alpha diversity in term-birth women only in PRJNA725416 (Fig. 2d). However, the *P* value for alpha diversity analysis in B/AA women of PRJNA725416 was less significant than in White women (Fig. 2a), and no taxa in B/AA women showed differential abundance linked to preterm birth in PRJNA725416. These results suggest inconsistent findings across cohorts for B/AA women, but PTB-associated differences were less pronounced in B/AA women compared to White and Asian women.

## Association between the vaginal microbiota and preterm birth mediated by age

The VMBs of different racial groups were further analyzed by stratifying women into age groups, with samples matched by BioProject, age, and gestational age (Fig. S6a for B/AA and Fig. S6b for White women). Interestingly, among B/AA women, the association between alpha diversity and PTB was not significant in the 17–25 years age group but became significant in the 25.1–32 years age group, with the strongest significance observed in those aged 32.1–42 years (Fig. 3a). The results were further confirmed by testing the association between alpha diversity and PTB using the inverse probability weighting procedure (see Materials and Methods and Data set S2). Differential abundance analysis (Fig. 3b) and composition bar plots (Fig. 3b) showed that term-birth VMBs in older B/AA women had more pronounced enrichment of health-associated *Lactobacillus* species, e.g., *L. gasseri* and *L. jensenii*, and a greater reduction in dysbiosis-associated taxa, e.g., *G. vaginalis* and *Prevotella amnii*. It was unexpected to find a lower abundance of protective *Lactobacillus* in term-birth women among B/AA individuals aged 17–25 years (Fig. 3b). However, this phenomenon was observed exclusively in the PRJNA393472 cohort (Fig. S7a through e) and not in the PRJNA725416 cohort (Fig. S7f through i).

Similarly, term-birth White women exhibited significantly lower alpha diversity in their VMBs, with this effect more pronounced in older age groups (Fig. 3d; Data set S2). Notably, VMBs in the 25.1–32 years age group had higher levels of *L. crispatus* and lower amounts of *L. gasseri* (Fig. 3e and f). Further analysis showed that VMBs dominated by over 50% *L. crispatus* had lower levels of alpha diversity compared to those dominated by over 50% *L. gasseri* or *L. jensenii* (Fig. S8). This finding partially explains the reduced alpha diversity observed in term-birth VMBs of White women aged 25.1–32 years and implies that *L. crispatus* may be more effective than *L. gasseri* or *L. jensenii* in maintaining VMB homeostasis. Consistent with our observation, a previous study has shown that *L. crispatus* has a lower level of l/d-lactic acid ratio, where d-lactic acid plays a more protective role for preterm birth (19).

Term-birth Asian women aged 32.1–42 years also showed significantly lower alpha diversity, along with enrichment of health-associated *Lactobacillus* species and a reduction in dysbiosis-associated taxa (Fig. S9). Due to the limited number of cases (fewer than 10 pairs), results of other age groups are not shown.

To further assess the consistency of our findings across different cohorts, we performed the same analysis on each independent cohort. Due to sample size limitations (fewer than 10 pairs), only two cohorts, i.e., PRJNA393472 and PRJNA725416, were included in this analysis. Observations from the PRJNA393472 cohort were fully consistent with the findings described above, as this cohort contributed the most samples to our study (Fig. S10a through f). The second-largest cohort with available maternal age data, PRJNA725416, had a limited sample size. Consequently, while the related results were not statistically significant, the observed trends aligned with our conclusions. More specifically, *L. crispatus* and *L. jensenii* were more abundant in term-birth cases than in

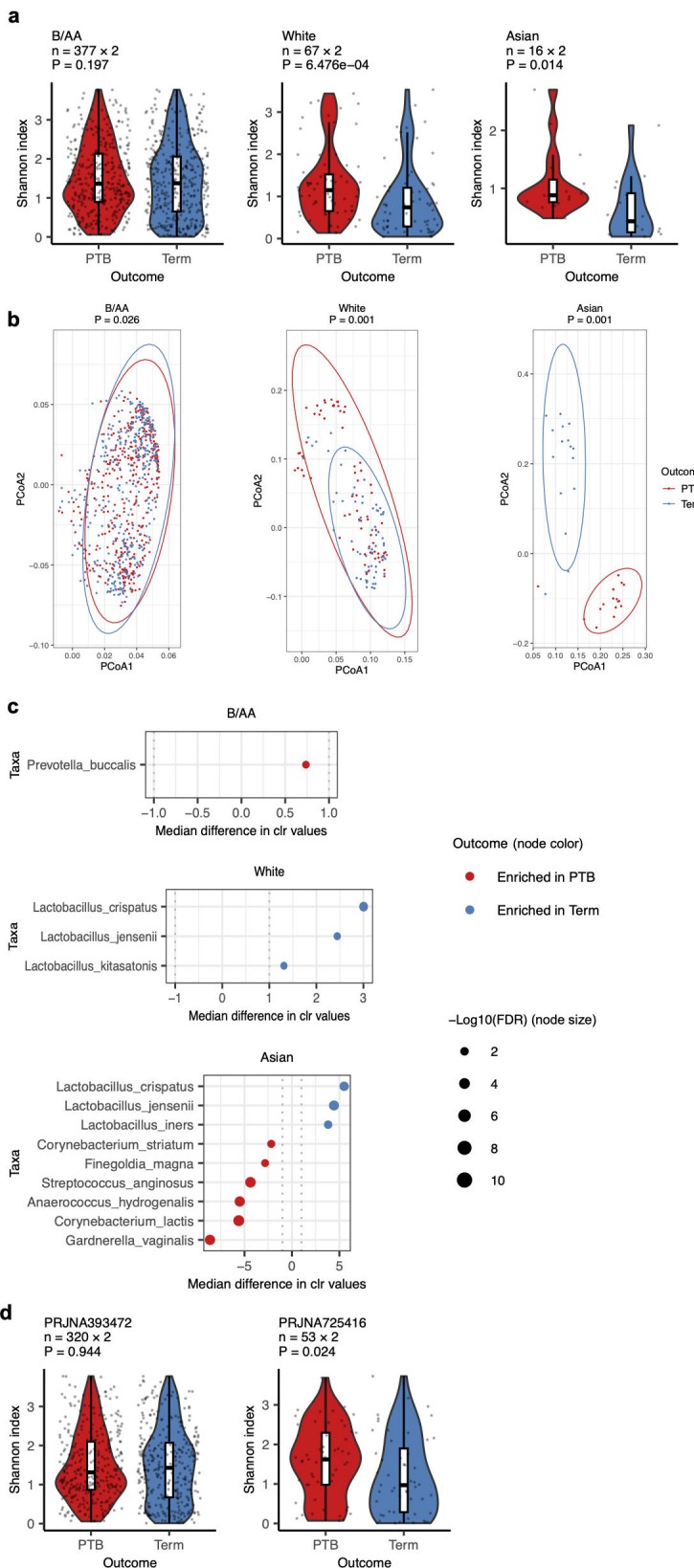

**FIG 2** The impact of race on the association between the VMB and PTB. The VMB samples were categorized into three racial subgroups, i.e., Black or African American, White, and Asian. In each

Fig 2 (Continued)

subgroup, PTB and term-birth samples were case-matched by BioProject, age, and gestational age (see Fig. S5). (a) Differences in alpha diversity between PTB and term birth within each racial group are shown. (b) Beta diversity differences visualized via PCoA are illustrated. (c) Differential abundances of taxa in the VMB of PTB and term birth in the three racial groups are visualized by dot plots. (d) Differences in alpha diversity between PTB and term birth in B/AA women in two individual cohorts are depicted. Statistical analyses were conducted using a two-sided Mann-Whitney U test for alpha diversity and the Adonis test for beta diversity. Changes in relative abundance were tested using the ALDEx2 package in R and quantified by the per-taxon median difference between conditions. False discovery rates (FDRs) were calculated with the Benjamini-Hochberg correction applied to the Mann-Whitney U test. Lines in the boxplots represent maximum, 75% quantile, median, 25% quantile, and minimum values from top to bottom.

PTB cases among older reproductive-age women (Fig. S10g through l). In contrast, dysbiosis-associated taxa, e.g., *G. vaginalis* and *Lachnospiraceae* BVAB1, exhibited a slower decline with age in PTB cases.

## Interaction between age and preterm birth in relation to vaginal bacterial taxa

Given the evident role of age in mediating the association between the VMB and PTB in all the studied races, mixed-effects models were constructed to evaluate how the interaction between age and PTB affects the relative abundance of VMB taxa using the case-matched data set as shown in Fig. S4d. The relative abundances were quantified using CLR normalized values. Results for the most abundant vaginal taxa are shown in Fig. 4, with the complete data set available in Data set S3.

The relative abundances of *L. crispatus* (Fig. 4a) and *L. jensenii* (Fig. 4b) increased with age in term-birth women but decreased in PTB women. Positive interaction coefficients between age and delivery outcome indicate that for each 1 unit increase in age, the relative abundances of *L. crispatus* and *L. jensenii* increased by 0.346 and 0.186 units more, respectively, in term births compared to PTBs. In contrast, the relative abundance of *L. gasseri* increased with age in both term and PTB women, with no significant difference in the rate of increase (Fig. 4c). For *L. iners*, *G. vaginalis*, and *Lachnospiraceae* BVAB1, abundance decreased with age (Fig. 4d through f). However, the decline in *L. iners* was slower in term births, while decreases in *G. vaginalis* and *Lachnospiraceae* BVAB1 were more pronounced in term births compared to PTBs. The results for *L. crispatus*, *L. jensenii*, *G. vaginalis*, and *Lachnospiraceae* BVAB1 suggest that the overall status of the VMB tends to be more optimal with age in term births but declines with age in PTB.

To examine the consistency of our findings across different cohorts, data from the PRJNA393472 and PRJNA725416 cohorts were tested independently. Similarly, observations from the PRJNA393472 cohort were fully consistent with the findings described above. While differences in the PRJNA725416 cohort were not statistically significant, the difference in the median values of Shannon index between PTB and term-birth cases was more pronounced in the 25–32 years age group than in the 17–25 years age group (Fig. S11), suggesting a similar trend.

## Association between the vaginal microbiota and preterm birth mediated by gestational age

The dbRDA revealed that race, age, and gestational age had a greater impact on the VMB than birth outcomes (Fig. 1c). The influence of race and age on the relationship between PTB and the VMB has also been investigated. Following this approach, VMBs were further stratified by gestational age and case-matched for PTB and term birth based on BioProject, race, age, and gestational age (Fig. S12).

Previous studies reported significantly lower alpha diversity in the VMB during early pregnancy, particularly in the first trimester, for term births compared to PTB (18, 20, 21).

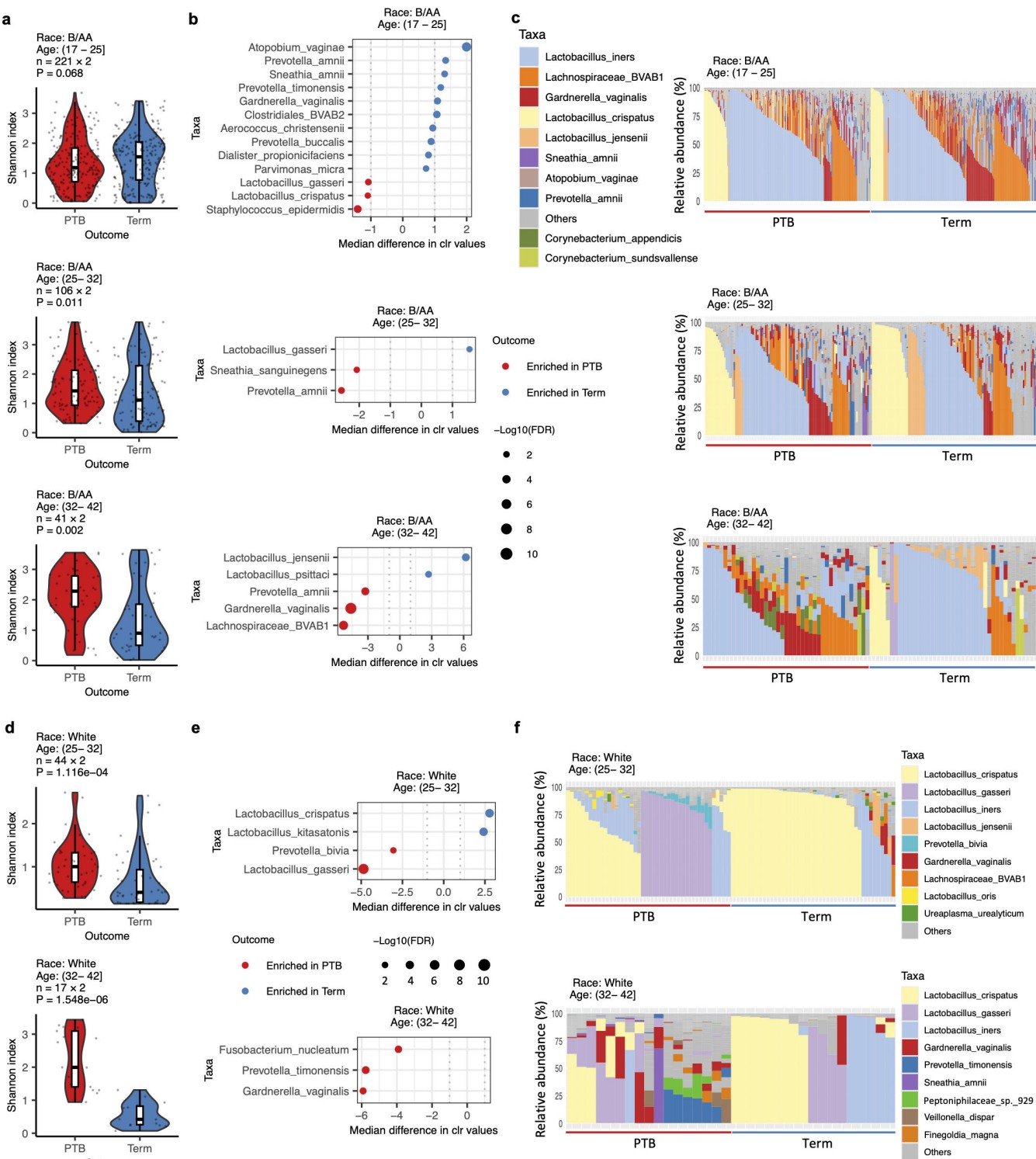

**FIG 3** The impact of age on the association between the VMB and PTB in different racial groups. The VMB samples were stratified by race and age. Within each subgroup, PTB and term-birth samples were case-matched based on BioProject, age, and gestational age (see Fig. S6). Differences in alpha diversity (a), differential taxa abundances (b), and VMB composition (c) between PTB and term birth in B/AA women across different age groups are shown. Differences in alpha diversity (d), differential taxa abundances (e), and VMB composition (f) between PTB and term birth in White women across different age groups are depicted. The results for Asian women aged 32.1–42 years are shown in Fig. S9. Subgroup analyses for other White and Asian women are excluded due to fewer than 10 sample pairs. Statistical analyses included the two-sided Mann-Whitney U test for alpha diversity, the Adonis test for beta diversity, and the ALDEx2 package in R for differential relative abundance analysis. Lines in the boxplots represent maximum, 75% quantile, median, 25% quantile, and minimum values from top to bottom.

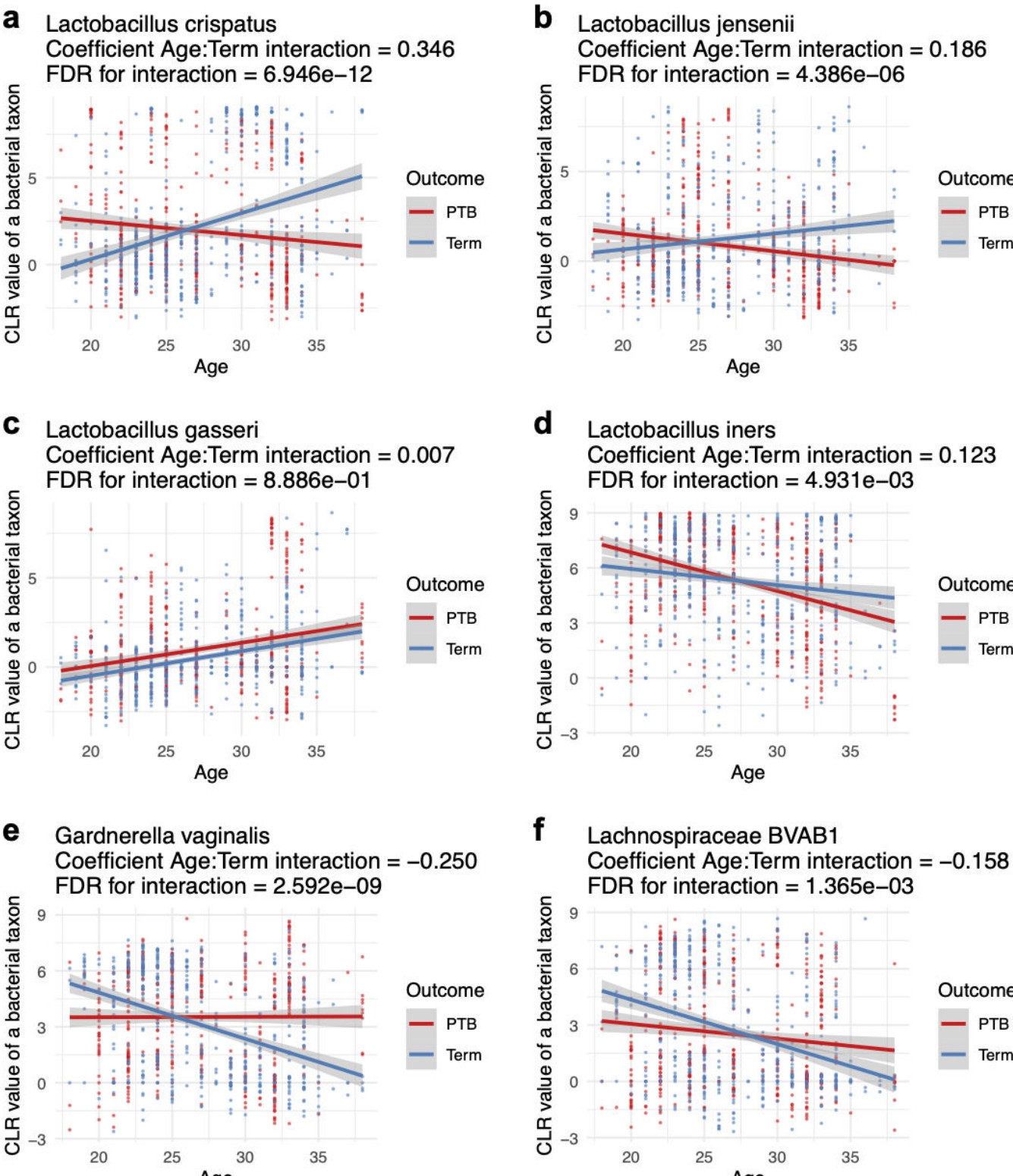

**FIG 4** Interaction between age and preterm birth in relation to vaginal bacterial taxa. A mixed-effects model was implemented using the lm function from the lme4 package to test the interaction between age and pregnancy outcome on a specific vaginal taxon. The coefficients representing the interaction, along with the FDRs of the interaction, for *L. crispatus* (a), *L. jensenii* (b), *L. gasseri* (c), *L. iners* (d), *G. vaginalis* (e), and *Lachnospiraceae* BVAB1 (f) are shown. A comprehensive list can be found in Data set S3.

Our analysis was consistent with these findings, showing reduced alpha diversity in term births compared to PTB, with a more pronounced difference in early pregnancy (*P* values for gestational age 44–135 days vs later gestational ages in both B/AA [Fig. 5a] and White [Fig. 5b] women). Unlike earlier studies that found no significant lower levels of alpha diversity in term births in late pregnancy (20, 21), our data indicated a lower alpha diversity in late pregnancy for term births, but only among older B/AA women (≥25.1 years, Fig. 5a). Furthermore, these results were confirmed by measuring the association between alpha diversity and PTB using the inverse probability weighting procedure (Data set S2). Due to limited cases among White and Asian participants, comparisons were restricted to White women aged 25.1–32 years. Since there were more pronounced differences between PTB and term births in White compared to B/AA women (Fig. 2), it was not surprising that term births also had lower levels of alpha diversity than PTB in White women aged 25.1–32 years.

## DISCUSSION

This study highlights the novel role of maternal age in mediating the association between the VMB and PTB. Age was identified as the second most influential factor affecting VMB composition, after race. Consistent with this finding, a previous study (17) identified age as the fourth most important factor influencing the VMB, following poverty level, education, and marital status, all of which are linked to race (10). Our data also showed that older women had more optimal VMB profiles, enriched in health-associated taxa, consistent with findings linking maternal age to a more optimal VMB (22). Interestingly, the interaction between age and PTB revealed that while the relative abundance of *L. crispatus* and *L. jensenii* increased with age in term births, it decreased in PTBs, indicating a more protective role of these vaginal taxa in older term-birth women in reproductive ages. Additionally, lower alpha diversity in the VMB of term pregnancies in late pregnancy was observed only in women aged 25.1 years or older. These findings highlight the complexity of age-related VMB changes and suggest that maternal age should be considered a critical factor in future studies on PTB and the VMB.

Previous studies have shown that estrogen is essential for increasing the abundance of protective *Lactobacillus* and maintaining overall vaginal health (1). However, studies have not found higher estrogen levels in women aged 25.1 years or older compared to those under 25 years (23, 24). Therefore, estrogen levels cannot explain the relationship between age and changes in the abundance of vaginal taxa. Additionally, there is currently no evidence to suggest that younger women experience more severe fluctuations in the composition of the VMB due to hormonal changes during reproductive years or that they possess a less effective immune system in maintaining VMB homeostasis. A plausible hypothesis is that age-related changes in behavioral and lifestyle factors may influence the VMB. For instance, more frequent sexual activity among younger women (25) could modulate the VMB, potentially resulting in a less stable or optimal VMB. However, the more pronounced differences in the VMB between PTB and term birth observed in older reproductive-age women could be attributed to a less efficient immune system in this age group. As immune system function declines with age (26), maintaining an optimal VMB may become more critical in reducing the risk of PTB in older women of reproductive age.

The difference in the VMB between PTB and term birth becomes notably less significant when cases are matched based on BioProject, race, age, and gestational age, compared to analyses without a case-matching design. Combined with the results of the dbRDA, this highlights the critical role of race, age, and gestational age in shaping the VMB. These factors should therefore be carefully accounted for in studies investigating associations between the VMB and PTB.

It is well established that an optimal VMB dominated by *Lactobacillus* species is associated with a lower risk of PTB. However, our data revealed a lower abundance of protective *Lactobacillus* in term-birth women within one specific group, B/AA aged 17–25. Notably, this observation was inconsistent across the two cohorts studied,

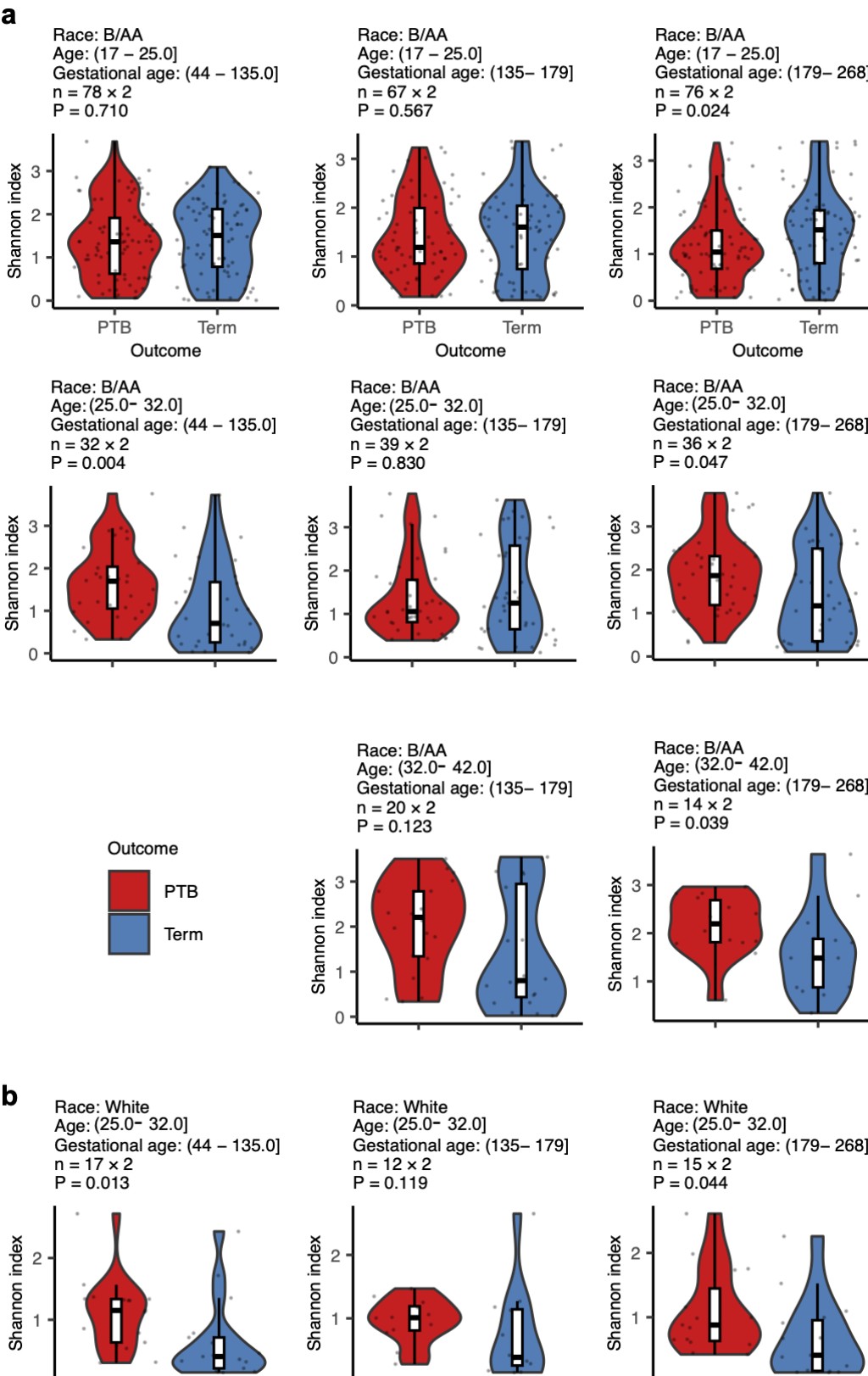

**FIG 5** The impact of gestational age on the association between the VMB and PTB in different racial and age groups. The VMB samples were stratified by race, age, and gestational age. Within each subgroup, PTB and term-birth samples were case-matched based on BioProject, age, and gestational age (see Fig. S12). Differences in alpha diversity between PTB and

Fig 5 (Continued)

term birth in B/AA (a) and White (b) women across different age and gestational age groups are shown. The two-sided Mann-Whitney U test was applied for the comparison between PTB and term birth. Subgroups with sample pairs fewer than 10 are not shown. Lines in the boxplots represent maximum, 75% quantile, median, 25% quantile, and minimum values from top to bottom.

PRJNA393472 and PRJNA725416, suggesting that additional factors may have influenced the results observed in the PRJNA393472 cohort for this subgroup. Due to limitations in the metadata available across all studied cohorts, this analysis included only three maternal characteristics: race, age, and gestational age. Incorporating more comprehensive metadata in future studies could offer a deeper understanding of the role of the VMB in PTB.

A significant limitation of this study is the disparity in case numbers across racial groups, which undermines the statistical power of certain subgroups. For example, Asian women across all ages and White women under 25 years had limited sample sizes. Although similar trends of differences between PTB and term birth were observed in these subgroups (data not shown), the robustness of statistical comparisons was compromised, making the results less reliable. Additionally, due to the limited sample size, cases of PTB and term birth could not be perfectly matched by age in a few comparisons, e.g., those shown in Fig. S5c and S6b. As a result, the observations in these specific comparisons may be influenced by age-related differences. However, in most comparisons in this study, case matching by age was well achieved, providing solid evidence to support the conclusions.

Another limitation is that the vaginal microbiome comprises not only bacteria but fungi and viruses, some of which have been associated with PTB (27–30). However, this study focuses on 16S rRNA sequencing data, limiting its scope to bacterial identification and excluding fungi and viruses present in the lower reproductive tract. Further research is needed to identify fungal and viral biomarkers in PTB and to explore their potential interactions with bacterial communities.

## ACKNOWLEDGMENTS

We acknowledge all project investigators from the University of Sheffield, Stanford University School of Medicine, Mayo Clinic, and Emory University for their contributions to generating the publicly shared data. We also thank the participants involved in sample collection for these BioProjects.

This work was supported by the Innovation Capability Strong Foundation Plan of Xi'an City (medical research project, 22YXYJ0117), Key Research and Development Program of Shaanxi (program no. 2023-YBSF-320), 2024 Shaanxi Provincial Health High-level Talent (team) Cultivation Plan—Young Talent Project, and the Fundamental Research Funds for the Central Universities (xzy012024146), National Natural Science Foundation of China (32170114 and 31770152), and Shaanxi Fundamental Science Research Project for Chemistry & Biology (program no. 22JHZ008).

Y.X.: funding acquisition, investigation, methodology, writing—review and editing. Q.W.: methodology, writing—original draft. D.L.: data curation, methodology. F.J.: data curation, methodology. F.C.: project administration, resources. Z.Z.: investigation, methodology. Y.S.: data curation, methodology. S.W.: project administration, supervision, funding acquisition, writing—review and editing. All authors read and approved the final manuscript.

## AUTHOR AFFILIATIONS

[1]Department of Laboratory Medicine, Northwest Women's and Children's Hospital, Xi'an, Shaanxi, China
[2]Department of Clinical Laboratory, Second Affiliated Hospital of Xi'an Jiaotong University, Xi'an, Shaanxi, China

[3]Shaanxi Institute of Microbiology, Xi'an, Shaanxi, China

[4]Key Laboratory of Resource Biology and Biotechnology in Western China, Ministry of Education, Provincial Key Laboratory of Biotechnology, College of Life Sciences, Northwest University, Xi'an, Shaanxi, China

## AUTHOR ORCIDs

Yun Xie  http://orcid.org/0009-0005-6147-9638
Fengan Jia  http://orcid.org/0009-0006-9179-6025
Fan Chang  http://orcid.org/0000-0002-6422-1852
Shiwei Wang  http://orcid.org/0000-0001-8890-9080

## FUNDING

| Funder | Grant(s) | Author(s) |
| --- | --- | --- |
| Innovation capability strong foundation plan of Xi'an City | 22YXYJ0117 | Shiwei Wang |
| Key Research and Development Program of Shaanxi | 2023-YBSF-320 | Shiwei Wang |
| 2024 Shaanxi Provincial Health high-level talent (team) cultivation plan-Young Talent Project | | Shiwei Wang |
| Fundamental Research Funds for the Central Universities | xzy012024146 | Shiwei Wang |
| National Natural Science Foundation of China | 32170114 | Shiwei Wang |
| National Natural Science Foundation of China | 31770152 | Shiwei Wang |
| Shaanxi Fundamental Science Research Project for Chemistry & Biology | 22JHZ008 | Shiwei Wang |

## AUTHOR CONTRIBUTIONS

Yun Xie, Conceptualization, Funding acquisition, Investigation, Methodology, Writing – original draft, Writing – review and editing | Qi Wang, Methodology, Writing – original draft | Dan Li, Data curation, Methodology | Fengan Jia, Data curation, Methodology | Fan Chang, Project administration, Resources | Zhen Zhang, Investigation, Methodology | Yanmei Sun, Data curation, Methodology | Shiwei Wang, Funding acquisition, Project administration, Supervision, Writing – original draft, Writing – review and editing

## DATA AVAILABILITY

Publicly available data sets were downloaded from the NCBI database (https://www.ncbi.nlm.nih.gov/). The BioProject IDs with data used in this study are as follows: PRJNA300860, PRJNA393472, PRJNA687274, and PRJNA725416. The underlying code for this study can be accessed at https://github.com/xyun1275/VMB_PTB/blob/main/results_visualization.R.

## ETHICS APPROVAL

Study protocols were approved by the Northwest Women's and Children's Hospital Institutional Review Board (IRB) under protocol number IRB 2024-100. Since all the publicly available data sets were obtained from the NCBI database (see details in "Data Availability"), information regarding informed consent to participate can be found in the original publications associated with these data sets (13–16). A STORMS checklist is available using the following link: https://github.com/xyun1275/VMB_PTB/blob/main/STORMS_Excel_1.03.xlsx.

## ADDITIONAL FILES

The following material is available online.

## Supplemental Material

**Data set S1 (mSystems00149-25-s0001.xlsx).** Coordinates of bacterial taxa on the dbRDA plot.

**Data set S2 (mSystems00149-25-s0002.xlsx).** Comparison of Shannon indices between PTB and term birth.

**Data set S3 (mSystems00149-25-s0003.xlsx).** Interaction between age and preterm birth in relation to vaginal bacterial taxa.

**Supplemental figures (mSystems00149-25-s0004.pdf).** Fig. S1 to S12.

## Open Peer Review

**PEER REVIEW HISTORY (review-history.pdf).** An accounting of the reviewer comments and feedback.

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
