## [Reviewer comments · mSystems]

Role of Age in Mediating the Association Between the Vaginal Microbiota and Preterm Birth

Yun Xie, Qi Wang, Dan Li, Fengan Jia, Fan Chang, Zhen Zhang, Yanmei Sun, and Shiwei Wang

Corresponding Author(s): Shiwei Wang, College of Life Science, Northwest University

Review Timeline:

Submission Date:	February 3, 2025
Editorial Decision:	March 19, 2025
Revision Received:	March 28, 2025
Accepted:	April 17, 2025

Editor: Shi Huang

Reviewer(s): The reviewers have opted to remain anonymous.

Transaction Report:

DOI: <https://doi.org/10.1128/msystems.00149-25>

Re: mSystems00149-25 (Role of Age in Mediating the Association Between the Vaginal Microbiota and Preterm Birth)

Dear Dr. Shiwei Wang:

The authors should carefully address the comments raised by reviewers on the sample size discrepancy, the impact of high maternal age, lack of validation, sample distribution bias in the subgroup analysis, etc.

Revision Guidelines

Sincerely,
Shi Huang
Editor
mSystems

Reviewer #1 (Comments for the Author):

In this manuscript, Xie and colleagues studied the impact of race, maternal age, and gestational age on the vaginal microbiota. Overall, this is a well-prepared manuscript, which highlights the essential role of maternal age in mediating the association between the vaginal microbiota and preterm birth. Since the impact of maternal age on the vaginal microbiota has not been well

studied in reproductive-age women, this study offers significant innovation. A few comments are listed below.

Major comments:

- 1 Line 115, since the total sample number is 2,366, why are there 3273 samples shown in Fig. S2d?
- 2 The authors demonstrate the influence of low and median maternal ages on the vaginal microbiome as shown on Fig. 1d. Could you please also elaborate on the impact of high maternal age on the vaginal microbiota?
- 3 Although the authors have conducted case-matching for the comparisons, some sample sets are not perfectly matched for age, as seen in Fig. S5c and S6b. Please discuss the potential impact of these imperfect case matches on the analysis.
- 4 Line 169, "a lower abundance of protective *Lactobacillus* in term birth women among B/AA individuals aged 17-25". The authors should further discuss the abnormal phenomenon. Is that caused by hormonal changes or by another factor?
- 5 Line 176, "This finding implies that *L. crispatus* may be more effective than *L. gasseri* or *L. jensenii* in maintaining VMB homeostasis." The interesting finding is consistent with an observation showing that *L. crispatus* has a lower level of l/d-Lactic acid ratio (<https://doi.org/10.1128/mbio.00460-13>), where d-lactic acid plays a more protective role for preterm birth.

Minor comments:

- 1 Line 34, Lachnospiraceae BVAB1 has been associated with BV and preterm birth, but this taxon is still not culturable. To my knowledge, there is not any identified virulence factor for this taxon. Thus, the authors should delete Lachnospiraceae BVAB1 in this sentence.
- 2 Line 65, please show the website link of the Sequence Read Archive.
- 3 Line 71, "The" should be changed to "the".
- 4 Line 97, does the term "Outcome" in the formula indicate preterm outcome? Please specify the definition.

Reviewer #2 (Comments for the Author):

This study investigates the relationship between the vaginal microbiome and preterm birth (PTB), leveraging a multi-cohort dataset with comprehensive stratification by race and age. It identifies *Lactobacillus*-dominated communities as protective, while dysbiotic communities are associated with increased PTB risk. The study also highlights maternal age as a significant factor shaping microbiome composition. Although the analytical approach is relatively simple, the study provides novel biological insights. However, several concerns need to be addressed.

Major comments:

1. A major limitation of this study is the absence of validation. While it integrates data from multiple cohorts, there is no independent dataset or external validation step to ensure that the observed associations between the vaginal microbiome and PTB are consistent across different populations. The authors should consider validating their findings using an independent cohort. Metagenomic profiling, functional annotation, and mechanistic studies could also be considered to further strengthen the study.
2. Another concern is the small sample sizes in certain subgroup analyses, particularly in race-stratified comparisons (e.g., Fig. 5b), which reduce statistical power and increase the risk of spurious associations. The authors should consider alternative approaches, such as inverse probability weighting (IPW), to better handle confounding in small subgroups.

Minor comments:

1. The study defines 25 and 32 years as age cutoffs but provides limited justification for these thresholds. It is unclear whether alternative cutoffs were tested or whether these thresholds are commonly used in vaginal microbiome research.
2. Some figures are not clearly presented. For instance, relative abundance plots (e.g., Fig. 2c) include too many colors, making interpretation difficult. The RDA plot (Fig. 1a) is also problematic, as the dots are large, non-transparent, and overlapping, while the layout and font sizes are inconsistent across subfigures, affecting readability. Improvements in figure clarity and consistency are necessary.
3. The vaginal microbiome includes both bacteria and fungi, yet this study relies solely on 16S rRNA sequencing, which does not capture fungal diversity. The authors should acknowledge this as a study limitation and discuss how it may impact their findings.

We sincerely appreciate the reviewers' thoughtful and insightful comments, which have greatly enhanced the quality of our manuscript. Our detailed point-by-point responses are provided below.

Reviewer #1 (Comments for the Author):

Major comments:

1 Line 115, since the total sample number is 2,366, why are there 3273 samples shown in Fig. S2d?

Response: Thanks for the kindly reminder. The number has been corrected to 2,266.

2 The authors demonstrate the influence of low and median maternal ages on the vaginal microbiome as shown on Fig. 1d. Could you please also elaborate on the impact of high maternal age on the vaginal microbiota?

Response: We understand the reviewer's concern. In the dbRDA analysis, only the two most important dimensions were displayed. The level for high maternal age was not shown because its contribution to the first two primary dimensions was weak, although it had some impact on less essential dimensions.

3 Although the authors have conducted case-matching for the comparisons, some sample sets are not perfectly matched for age, as seen in Fig. S5c and S6b. Please discuss the potential impact of these imperfect case matches on the analysis.

Response: We agree with the reviewers comments and related discussion has been added in the new version at lines 264-267 [Additionally, due to the limited sample size, cases of PTB and term birth could not be perfectly matched by age in a few comparisons, e.g., those shown in Fig. S5c and S6b. As a result, the observations in these specific comparisons may be influenced by age-related differences. However, in most comparisons in this study, case matching by age was well achieved, providing solid evidence to support the conclusions.]

4 Line 169, "a lower abundance of protective *Lactobacillus* in term birth women among B/AA individuals aged 17-25". The authors should further discuss the abnormal phenomenon. Is that caused by hormonal changes or by another factor?

Response: We appreciate the reviewer's comment. The related discussion has been presented at lines 254-260 [It is well established that an optimal VMB dominated by *Lactobacillus* species is associated with a lower risk of PTB. However, our data revealed a lower abundance of protective *Lactobacillus* in term-birth women within one specific group, B/AA aged 17-25. Notably, this observation was inconsistent across the two cohorts studied, PRJNA393472 and PRJNA725416, suggesting that additional factors may have influenced the results observed in the PRJNA393472 cohort for this subgroup. Due to limitations in the metadata available across all studied cohorts, this analysis included only three maternal characteristics: race, age, and gestational age. Incorporating more comprehensive metadata in future studies could offer a deeper understanding of the role of the VMB in PTB.]

5 Line 176, "This finding implies that *L. crispatus* may be more effective than *L. gasseri* or *L. jensenii* in maintaining VMB homeostasis." The interesting finding is consistent with an observation showing that *L. crispatus* has a lower level of l/d-Lactic acid ratio (<https://doi.org/10.1128/mbio.00460-13>), where d-lactic acid plays a more protective role for preterm birth.

Response: Thanks for the insightful comment. A new sentence as well as the referred citation has been added at line 181 [Consistent with our observation, a previous study has shown that *L. crispatus* has a lower level of l/d-Lactic acid ratio, where d-lactic acid plays a more protective role for preterm birth⁴.]

Minor comments:

1 Line 34, Lachnospiraceae BVAB1 has been associated with BV and preterm birth, but this taxon is still not culturable. To my knowledge, there is not any identified virulence factor for this taxon. Thus, the authors should delete Lachnospiraceae BVAB1 in this sentence.

Response: Thanks for the careful review. This sentence at line 29 has been corrected by the deletion of BVAB1. [In contrast, many non-*Lactobacillus* species, such as *Gardnerella vaginalis* and *Sneathia amnii*, produce virulence factors and are linked to various diseases in the female lower reproductive tract.]

2 Line 65, please show the website link of the Sequence Read Archive.

Response: The link to the website has been added at line 60 [Raw 16S rRNA sequencing data were publicly available for download from the Sequence Read Archive (SRA, <https://www.ncbi.nlm.nih.gov/sra>)].

3 Line 71, "The" should be changed to "the".

Response: Corrected, thanks.

4 Line 97, does the term "Outcome" in the formula indicate preterm outcome? Please specify the definition.

Response: Thanks for the reminder. The sentence is revised to “A mixed-effects model was implemented using the lm function from the lme4 package, with the formula specified as Bacterium ~ Age * pregnancy outcome” at line 97.

Reviewer #2 (Comments for the Author):

Major comments:

1. A major limitation of this study is the absence of validation. While it integrates data from multiple cohorts, there is no independent dataset or external validation step to ensure that the observed associations between the vaginal microbiome and PTB are consistent across different populations. The authors should consider validating their findings using an independent cohort. Metagenomic profiling, functional annotation, and mechanistic studies could also be considered to further strengthen the study.

Response: We appreciate the reviewer's insightful comment. The key finding of our study is the association between maternal age and preterm birth (PTB). Despite our best efforts, we were unable to identify an additional independent cohort for validation, as most available cohorts either have limited case numbers or lack information on maternal age. To address whether this finding is consistent across different cohorts, we analyzed the included cohorts independently. As shown in the newly added figures (Fig. S10 and S11), results from the PRJNA393472 cohort are fully consistent with our findings, as this cohort contributes the most samples to our study. The second-largest cohort with available maternal age data, PRJNA725416, has a limited sample size. Consequently, while the related results are not statistically significant, the observed trends align with our conclusions. Specifically, we found that *Lactobacillus crispatus* and *Lactobacillus jensenii* were more abundant in term birth

cases than in PTB cases among older reproductive-age women (Fig. S11 g-l). In contrast, dysbiosis-associated taxa, e.g., *Gardnerella vaginalis* and *Lachnospiraceae* BVAB1, exhibited a slower decline with age in PTB cases. Additionally, the difference in the median value of Shannon index between PTB and term birth cases was more pronounced in the (25–32] age group than in the (17–25] age group (Fig. S10b). The corresponding text has been added at lines 186-193. [To further assess the consistency of our findings across different cohorts, we performed the same analysis on each independent cohort. Due to sample size limitations (fewer than 10 pairs), only two cohorts, i.e., PRJNA393472 and PRJNA725416, were included in this analysis. Observations from the PRJNA393472 cohort were fully consistent with the findings described above, as this cohort contributed the most samples to our study (Fig. Sxxx a-f). The second-largest cohort with available maternal age data, PRJNA725416, had a limited sample size. Consequently, while the related results were not statistically significant, the observed trends aligned with our conclusions. More specifically, *L. crispatus* and *L. jensenii* were more abundant in term birth cases than in PTB cases among older reproductive-age women (Fig. Sxxx g-l). In contrast, dysbiosis-associated taxa, e.g., *G. vaginalis* and *Lachnospiraceae* BVAB1, exhibited a slower decline with age in PTB cases.] and lines 208-212. [To examine the consistency of our findings across different cohorts, data from the PRJNA393472 and PRJNA725416 cohorts were tested independently. Similarly, observations from the PRJNA393472 cohort were fully consistent with the findings described above. While differences in the PRJNA725416 cohort were not statistically significant, the difference in the median values of Shannon index between PTB and term birth cases was more pronounced in the 25–32 age group than in the 17–25 age group (Fig. Sxxx b), suggesting a similar trend.] Furthermore, we totally agree with the reviewer that metagenomic profiling and a study at gene and pathway levels will lead to the discovery on mechanisms. We are collecting our own cohort for further validation as suggested by the reviewer.

2. Another concern is the small sample sizes in certain subgroup analyses, particularly in race-stratified comparisons (e.g., Fig. 5b), which reduce statistical power and increase the risk of spurious associations. The authors should consider alternative approaches, such as inverse probability weighting (IPW), to better handle confounding in small subgroups.

Response: We appreciate the reviewer's great comment. All comparisons of the Shannon index between PTB and term birth were conducted using values adjusted by the inverse probability weighting procedure, and the results remain consistent with our current conclusions. For details, please refer to lines 82-88 [Alternatively, in case small sample sizes in certain subgroup analyses increase the risk of spurious associations, Shannon index was adjusted by the inverse probability weighting procedure. More specifically, propensity scores were estimated using a logistic regression model with the glm function in R, where the probability of the outcome (PTB vs. term birth) was predicted based on potential confounders, i.e., gestational age at collection and maternal age. Next, inverse probability weights were computed using these scores to balance the data. The weights were applied in the weighted regression using the svyglm function of the survey package in R. This approach ensured that the association between the Shannon index and pregnancy outcome was not biased by confounding factors.], lines 166-169 [Interestingly, among B/AA women, the association between alpha diversity and PTB was not significant in the 17-25 age group but became significant in the 25.1-32 age group, with the strongest significance observed in those aged 32.1-42 (Fig. 3a). The results were further confirmed by testing the association between alpha diversity and PTB using the inverse probability weighting procedure (see Methods and Supplementary Dataset 2).], and line 223 [Furthermore, these results were confirmed by measuring the association between alpha diversity and PTB using the inverse probability weighting procedure (Supplementary Dataset 1).] as well as the new Supplementary Dataset 2.

Minor comments:

1. The study defines 25 and 32 years as age cutoffs but provides limited justification for these

thresholds. It is unclear whether alternative cutoffs were tested or whether these thresholds are commonly used in vaginal microbiome research.

Response: Thanks for the reminder. To make it more clear, the sentence at line 128 were modified. [For comparability, age (Fig. 1a) and gestational age (Fig. 1b) were divided into three categories with similar case numbers to achieve a comparable degree of freedom to race and pregnancy outcome in the dbRDA analysis. This approach also helps prevent bias in group size distribution.]

2. Some figures are not clearly presented. For instance, relative abundance plots (e.g., Fig. 2c) include too many colors, making interpretation difficult. The RDA plot (Fig. 1a) is also problematic, as the dots are large, non-transparent, and overlapping, while the layout and font sizes are inconsistent across subfigures, affecting readability. Improvements in figure clarity and consistency are necessary.

Response: Thank you for pointing out these issues. We have reorganized Fig. 2c and added annotations, i.e., “(node color)” and “(node size)”, to the legend for better clarity. For the RDA plot, we adjusted the transparency of the dots to 50%. However, the dots (or bacterial taxa) at the center remain overlapped because they have minimal associations with impact factors involved in this analysis, e.g., race and age. As a result, the RDA plot primarily highlights taxa with strong associations, such as *L. crispatus*, *L. iners*, and *G. vaginalis*. In response to the comment, we have added a new Supplementary Dataset 1 to provide the detailed values of each taxon on the RDA plot. Additionally, a sentence has been included in the figure 1 legend [Detailed coordinates of bacterial taxa on the dbRDA plot are listed in Supplementary Dataset 1.]

3. The vaginal microbiome includes both bacteria and fungi, yet this study relies solely on 16S rRNA sequencing, which does not capture fungal diversity. The authors should acknowledge this as a study limitation and discuss how it may impact their findings.

Response: Appreciate for the comment. A new paragraph has been added in the discussion to discuss fungi in the reproductive tract. Please see lines 268-271 [Another limitation is that the vaginal microbiome comprises not only bacteria but fungi and viruses, some of which have been associated with PTB²²⁻²³. However, this study focuses on 16S rRNA sequencing data, limiting its scope to bacterial identification and excluding fungi and viruses present in the lower reproductive tract. Further research is needed to identify fungal and viral biomarkers in PTB and to explore their potential interactions with bacterial communities.

]

Re: mSystems00149-25R1 (Role of Age in Mediating the Association Between the Vaginal Microbiota and Preterm Birth)

Dear Dr. Shiwei Wang:

Your manuscript has been accepted, and I am forwarding it to the ASM production staff for publication. Your paper will first be checked to make sure all elements meet the technical requirements. ASM staff will contact you if anything needs to be revised before copyediting and production can begin. Otherwise, you will be notified when your proofs are ready to be viewed.

Sincerely,
Shi Huang
Editor
mSystems

Reviewer #1 (Comments for the Author):

All my questions have been appropriately addressed in the new version.

Reviewer #2 (Comments for the Author):

I appreciate the authors' thoughtful and thorough revisions. The manuscript is now well-prepared for publication. One minor suggestion for improvement would still be the aesthetic quality of the figures. I appreciate the added annotations. Further improvements in layout, font consistency, and spacing would be preferred but not necessary.